# OOD Learner via In-Context Learning

## Abstract

Out-of-distribution (OOD) detectors are built on classifier to identify test samples that do not belong to any of their training classes. For classifiers based on pretrained vision-language models (VLMs), recent methods construct OOD detectors using text and few-shot in-distribution (ID) images. In this work, we introduce a versatile framework for few-shot OOD detection through in-context learning (ICL). Instead of building an OOD detector for specific ID datasets, we propose a universal **OOD Learner** that can adapt to arbitrary ID datasets using few-shot texts and images as context, without the need for fine-tuning. Our method is implemented as an attention-based module and pretrained on pseudo-class data curated from large-scale text-image pairs. Experiments demonstrate that our framework achieves SOTA performance and efficiency in few-shot OOD detection.

## 1 Introduction

Out-of-distribution (OOD) detection (Berend et al., 2020; Yang et al., 2024) aims to identify data points that deviate from a given in-distribution (ID) dataset. This is particularly crucial in high-stake applications such as healthcare, autonomous driving, and finance, where unexpected inputs can lead to errors or security breaches. Effective OOD detection enables models to recognize unfamiliar data, allowing them to withhold predictions or seek further verification.

Earlier works (Hendrycks & Gimpel, 2016; Liang et al., 2017; Lee et al., 2017; Liu et al., 2020; Chen et al., 2021; Sun et al., 2022; Ming et al., 2022b; Wang et al., 2022) have primarily focused on OOD detection for classification models trained on closed-set data with a fixed number of classes. The emergence of pretrained large vision-language models (VLMs), such as CLIP (Radford et al., 2021), has reshaped both classifier design and OOD detection, enabling OOD detectors to be built with only a few image examples or even solely from text descriptions. MCM (Ming et al., 2022a) was the first study to demonstrate that CLIP can inherently function as an OOD classifier. It achieves so by using CLIP features from simple text descriptions, such as "*a photo of a dog*," as the class representation for OOD detection.

Three major directions have merged to improve CLIP-based OOD detection. The first direction (Jiang et al., 2024; Ding & Pang, 2024; Cao et al., 2024) focuses on text prompt augmentation by incorporating negative prompts collected from large text corpus or generated by large language models (LLMs). The second direction (Wang et al., 2023; Esmaeilpour et al., 2022) explores training auxiliary modules to refine OOD detection. For instance, CLIPN (Wang et al., 2023) introduces an additional text encoder to align images with text containing the concept of "*no*." These two approaches rely solely on text for OOD detection, making them part of "zero-shot" OOD detection. In contrast, the third direction (Zhou et al., 2022; Miyai et al., 2024; Bai et al., 2024; Li et al., 2024) leverages a flexible number of images (*i.e.*, "few-shot" OOD detection) to refine text prompts while keeping CLIP frozen, a technique known as "prompt tuning." Specifically, these methods introduce a trainable prompt vector as a shared prefix of text prompts, which is tuned to improve text-image alignment with corresponding images.

Although prompt tuning methods improve OOD detection over zero-shot methods, they have several limitations. First, they exhibit limited generalization and require separate fine-tuning for each ID dataset. As shown in Sec. 4.2, a prompt vector tuned on ImageNet-1K performs poorly when applied to Texture (Cimpoi et al., 2014). Second, prompt tuning lacks flexibility to new information, such as additional shots of images or new ID classes. As the number of ID few-shot images and ID classes increases, those methods require retuning with the entire data. Finally, because prompt tuning relies on classification loss, it is ineffective when the ID dataset contains only a single class.

Motivated by these limitations, this work explores the following question:

*Can models learn a universal OOD concept that generalizes across datasets without extra tuning?*

A universal OOD model should estimate the underlying distribution from a few ID examples and detect anything outside it. We term it "**OOD Learner**." GPT-3 (Brown et al., 2020) shows significant performance gains across tasks (Bisk et al., 2020; Wang et al., 2019) when given task-specific few-shot examples as context, an ability known as in-context learning (ICL), eliminating the need for fine-tuning. Inspired by this, we propose an ICL framework for OOD Learner, as shown in Fig. 1, that incorporates a variable number of text and image examples from arbitrary classes to learn ID representations for OOD detection. To enable such flexibility, we propose an attention-based module with separate text and image channels. Within each channel, self-attention allows information exchange within each modality, while cross-attention bridges the modality gap (Liang et al., 2022) by integrating text and image features. Finally, ID representations are aggregated from the outputs of attention modules and compared with test images to compute OOD scores.

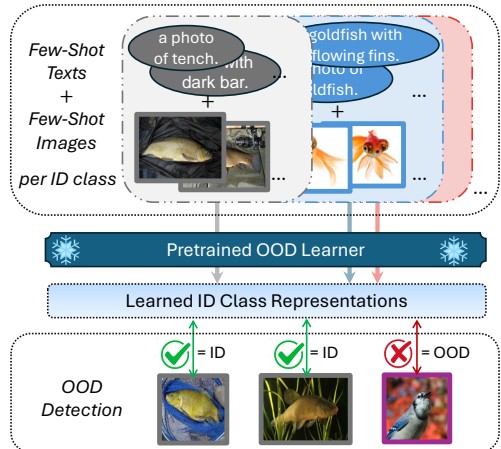

Figure 1: **OOD Learner.** Illustrated with multiple fish classes as ID classes, OOD Learner takes few-shot texts and images as input to learn ID class representations. These representations are then used to compare against test images for OOD detection. This process is finetuning-free, distinguishing OOD Learner from existing CLIP-based OOD detection methods.

The universal OOD concept allows our OOD Learner to adapt to any ID dataset without finetuning, unlike prompt tuning-based few-shot methods, as shown in Fig. 2. A key factor in pretraining the learner is leveraging a large, diverse data corpus covering a wide range of classes. Instead of using standard closed-set datasets such as `ImageNet` (Deng et al., 2009), we curate a corpus from large-scale text-image datasets `LAION-400M` (Schuhmann et al., 2021). We generate numerous pseudo-classes by applying ANNOY (Indyk & Motwani, 1998) to CLIP features to cluster images with high similarity to a given text[1]. Using just $10\%$ of `LAION-400M`, we obtain $\sim$30M pseudo-classes, which far exceed 1K from `ImageNet`). Our method achieves state-of-the-art performance and efficiency on OOD detection benchmarks. Further evaluations on varying ID datasets and ablation studies demonstrate its adaptability, whereas prompt tuning-based methods require additional fine-tuning. Our contributions are summarized as follows:

- We propose an OOD Learner, a novel ICL-based framework for OOD detection that adapts to arbitrary ID datasets and versatile scenarios without fine-tuning.

- We introduce an effective data curation strategy to construct large-scale pseudo-classes from text-image pairs such as `LAION-400M` (Schuhmann et al., 2021) for pretraining the OOD Learner.

- We demonstrate that the OOD Learner outperforms CLIP-based few-shot OOD detection methods in both SOTA performance and efficiency. Unlike prompt-tuning methods, OOD Learner generalizes across different ID datasets without requiring fine-tuning.

## 2 RELATED WORK

**Zero-Shot OOD Detection.** MCM (Ming et al., 2022a) is the first work to leverage a pretrained CLIP (Radford et al., 2021) model with class labels (*e.g.*, "dog" and "cat") for OOD detection, outperforming traditional classifier-based OOD detection methods (Hendrycks & Gimpel, 2016; Liang et al., 2017; Lee et al., 2017; Liu et al., 2020). Performing OOD detection by simply leveraging class names as text prompts for CLIP is referred to as "zero-shot OOD detection." Following MCM,

---

[1]Clarification: though concept-wise our framework extends to multiple texts and images, in our experiments, we use one text + multiple images for training and testing to follow the literature (1 text + 16 images).

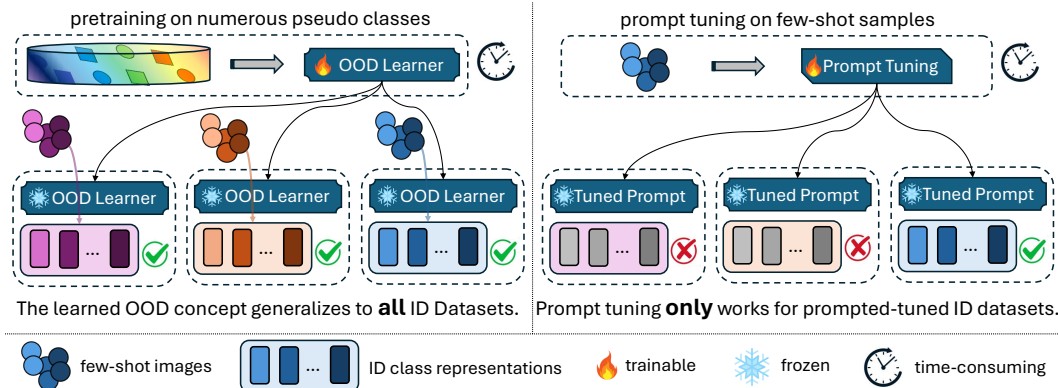

Figure 2: **OOD Learner vs. prompt tuning.** Unlike existing CLIP-based few-shot OOD detection methods that use few-shot samples for prompt tuning, OOD Learner is pretrained on numerous pseudo-classes and directly learn in-distribution (ID) representations from few-shot inputs. This allows adaption to arbitrary ID datasets without fine-tuning. (Color indicates different classes.)

two main approaches further enhance zero-shot OOD detection. The first approach considers class labels as positive prompts and introduces additional negative prompts. For instance, NegLabel (Jiang et al., 2024) and CLIP-OLE (Ding & Pang, 2024) construct extra negative labels—*i.e.*, negative prompts—from a large text corpus. Alternatively, EOE (Cao et al., 2024) uses LLMs to generate negative labels. The second approach leverages additional pretraining datasets to train extra models. For example, CLIPN (Wang et al., 2023) utilizes CC3M (Sharma et al., 2018a) to pretrain an additional text encoder specifically for the "no" concept, and then calibrates text features from both the additional text encoder and CLIP's original text encoder for OOD detection. ZOC (Esmaeilpour et al., 2022) employs MS-COCO (Lin et al., 2014) to finetune a text-based image description generator, which is then used to generate unknown class names, *i.e.*, negative labels.

**CLIP-Based Few-Shot OOD Detection.** When provided with both class names and few-shot images per class from the ID dataset, one can perform "few-shot OOD detection" (Zhou et al., 2022; Miyai et al., 2024; Li et al., 2024; Bai et al., 2024; Lafon et al., 2024; Fu et al., 2024; Chen et al., 2024) on CLIP. Existing methods rely heavily on prompt tuning and often depend on carefully utilizing the few-shot images. CoOp (Zhou et al., 2022) is the pioneering work to apply prompt tuning to CLIP, initially designed for classification but later shown effective for OOD detection (Miyai et al., 2024). This method includes two stages as shown in Fig. 2: (i) add a trainable prompt vector as a shared prefix to text prompts of ID classes, and tune this prefix so that the tuned prompt prefix + text prompt of each ID class is aligned with in-class images; (ii) the CLIP feature of the tuned prompt prefix + text prompt of each ID class is regarded as ID class representation for that class, and these representations are further used to compare with test images for OOD detection. LoCoOp (Miyai et al., 2024) further extends CoOp by incorporating local feature extraction. Specifically, LoCoOp considers local features of individual image regions, selecting the top $K$ features most similar to the text embedding to compute image-text similarity. Moreover, background features serve as outlier features, contributing to a regularization term in the training loss. Beyond prompt tuning, the idea of negative labels has been adapted from zero-shot to few-shot settings (Li et al., 2024; Bai et al., 2024). Building on CoOp, NegPrompt (Li et al., 2024) introduces negative prompts with additional loss terms for tuning and regularization, showing that these prompts can generalize across domains. ID-Like (Bai et al., 2024) crops few-shot images into object patches (containing the main object) and object-like patches (containing minimal or no object content). ID-Like then performs prompt tuning to align positive prompts with object patches and aligns negative prompts with object-like patches. In contrast to these prompt tuning-based methods, this paper introduces OOD Learner that performs few-shot OOD detection without requiring any fine-tuning.

**ICL.** First observed in GPT-3 (Brown et al., 2020), ICL is an emergent ability of LLMs to adapt to new tasks without parameter updates, relying solely on examples provided within the prompt. This demonstrates that a pretrained model can perform diverse downstream tasks without additional training. Inspired by ICL, we introduce OOD Learner that generalizes across diverse OOD scenarios with context of examples to represent ID distributions.

# 3 METHOD

## 3.1 PRELIMINARY

We introduce an **OOD Learner**, a model that generalizes OOD detection across diverse distributions without relyingper-dataset fine-tuning. The OOD Learner is designed to adapt to new ID distributions with just a few examples. Formally, we perform OOD detection over a distribution over $C$ in-distribution (ID) image classes, denoted as $\mathcal{C} = \{c^{(1)}, c^{(2)}, \ldots, c^{(C)}\}$ (see Appendix C for notation details). Each ID class is represented by a text description $\mathbf{t}^c$, such as "*a photo of a dog*," and $K$-shot images $\mathcal{I}^c = \{\mathbf{i}_1^c, \ldots, \mathbf{i}_K^c\}$. Given the ID context $\{\mathbf{t}^c, \mathcal{I}^c\}_{c=1}^C$ and a query image $\mathbf{i}$, the goal of ICL is to define a mapping function that enables few-shot OOD detection without fine-tuning:

$$f(\ \cdot\ |\{\mathbf{t}^c, \mathcal{I}^c\}_{c=1}^C) : \mathbf{i} \to [0, 1], \tag{1}$$

where the output is expected to be close to $1$ if the image $\mathbf{i}$ belongs to the ID classes and $0$ otherwise. A threshold is applied to convert the score into a binary indicator to determine ID or OOD.

We design the OOD Learner as $g(\cdot)$, which takes text and images from an ID class as input and outputs the class representation: $\boldsymbol{g}^c = g(\mathbf{t}^c, \mathcal{I}^c)$. Combined with the objective in Eq. (1), the OOD score for a test image $\mathbf{i}$ is defined as:

$$f(\mathbf{i}|\{\mathbf{t}^c, \mathcal{I}^c\}_{c=1}^C) = \max_{d \in \mathcal{C}} \left\{ \frac{\text{sim}(\boldsymbol{i}, \boldsymbol{g}^d)}{\sum_{c \in \mathcal{C}} \text{sim}(\boldsymbol{i}, \boldsymbol{g}^c)} \right\}, \tag{2}$$

where $\boldsymbol{i}$ represents the CLIP feature of $\mathbf{i}$ (with $\boldsymbol{i}$ and $\boldsymbol{t}$ denoting the CLIP features of image $\mathbf{i}$ and text $\mathbf{t}$ respectively), and $\text{sim}(\cdot, \cdot)$ is the cosine similarity function. The objective in Eq. (2) can be extended to Q-shot texts $\mathcal{T}^c = \{\mathbf{t}_1^c, \ldots, \mathbf{t}_Q^c\}$ by allowing the OOD Learner $g(\cdot)$ to take both multiple texts and images as input.

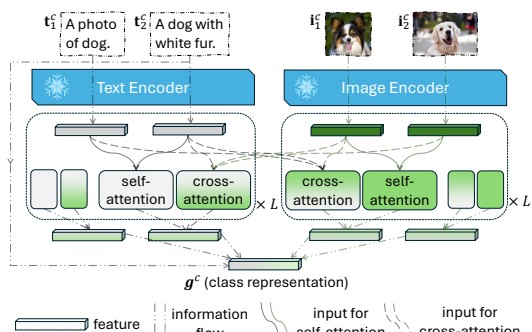

## 3.2 ATTENTION-BASED OOD LEARNER

This section details the OOD Learner $g(\cdot)$, an attention-based network that combines text and images into a class representation. As shown in Fig. 3, the network extracts text and image features using frozen CLIP encoders, processes them through $L$-layer attention modules, and uses a final aggregation layer to generate $\boldsymbol{g}^c$.

Figure 3: **Attention-based module.** An illustration with 2 texts, $\mathbf{t}_1^c$ ("*A photo of a dog*") and $\mathbf{t}_2^c$ ("*A dog with white fur*"), and 2 images $\mathbf{i}_1^c$ (a dog) and $\mathbf{i}_2^c$ (another dog) as model inputs.

**Attention module.** The proposed attention module consists of two separate channels for image and text, and thus can handle varying numbers of texts and images. The self-attention module enables information exchange within each modality, while the cross-attention module facilitates interaction between the two. Given input features $\{\boldsymbol{x}_1, \ldots, \boldsymbol{x}_K\}$, a self-attention module computes the output features $\{\boldsymbol{y}_1, \ldots, \boldsymbol{y}_K\}$ as follows:

$$\forall i, \ \boldsymbol{q}_i = \boldsymbol{W}_{\text{query}} \boldsymbol{x}_i, \ \boldsymbol{k}_i = \boldsymbol{W}_{\text{key}} \boldsymbol{x}_i, \ \boldsymbol{v}_i = \boldsymbol{W}_{\text{value}} \boldsymbol{x}_i, \tag{3}$$

$$\forall i, \ \boldsymbol{h}_i = \frac{\sum_{j=1}^K \boldsymbol{v}_j \cdot \langle \boldsymbol{q}_i, \boldsymbol{k}_j \rangle}{\sum_{j=1}^K \langle \boldsymbol{q}_i, \boldsymbol{k}_j \rangle}, \ \boldsymbol{y}_i = \text{MLP}(\boldsymbol{h}_i). \tag{4}$$

Here, $\boldsymbol{W}_{\text{query}}, \boldsymbol{W}_{\text{key}}, \boldsymbol{W}_{\text{value}}$ are trainable matrices mapping an input feature to key, query, and value. Then, $\boldsymbol{h}_i$ is computed to aggregate information for any input size $K$. Finally, an MLP layer is applied to transfer the aggregation to output $y_i$. Similarly, for the source features $\{\boldsymbol{x}_1^s, \ldots, \boldsymbol{x}_K^s\}$ to gather information from target features $\{\boldsymbol{x}_1^t, \ldots, \boldsymbol{x}_Q^t\}$, cross-attention performs the following computation:

$$\forall i, \ \boldsymbol{q}_i^s = \boldsymbol{W}_{\text{query}}^s \boldsymbol{x}_i^s, \ \boldsymbol{k}_i^t = \boldsymbol{W}_{\text{key}}^t \boldsymbol{x}_i^t, \ \boldsymbol{v}_i^t = \boldsymbol{W}_{\text{value}}^t \boldsymbol{x}_i^t, \tag{5}$$

$$\forall i, \ \boldsymbol{h}_i^s = \frac{\sum_{j=1}^Q \boldsymbol{v}_j^t \cdot \langle \boldsymbol{q}_i^s, \boldsymbol{k}_j^t \rangle}{\sum_{j=1}^Q \langle \boldsymbol{q}_i^s, \boldsymbol{k}_j^t \rangle}, \ \boldsymbol{y}_i^s = \text{MLP}^s(\boldsymbol{h}_i^s). \tag{6}$$

Thanks to the attention mechanism (Bahdanau, 2014; Vaswani et al., 2017), which computes a weighted sum over features with arbitrary size $K$ (as shown in Eqs. (4) and (6)), OOD Learner can process flexible-shot texts and images through self- and cross-attention. While we describe the attention mechanism to illustrate how a variable number of inputs is handled by the same attention module, our implementation employs multi-head attention (Vaswani et al., 2017), where we define $\boldsymbol{x}_i, \boldsymbol{y}_i, \boldsymbol{q}_i, \boldsymbol{k}_i, \boldsymbol{v}_i, \boldsymbol{h}_i \in \mathbb{R}^d$, with $d$ as the hidden dimension of the attention module.

**Final aggregation layer.** Let the outputs of the last attention be $\{\tilde{\boldsymbol{t}}_1^c, ..., \tilde{\boldsymbol{t}}_Q^c\}$ for texts and $\{\tilde{\boldsymbol{i}}_1^c, ..., \tilde{\boldsymbol{i}}_K^c\}$ for images. We aggregate them in the final layer as $\boldsymbol{g}^c = \text{norm}(\sum_{q=1}^Q \boldsymbol{t}_q^c + \lambda \cdot \sum_{k=1}^K \tilde{\boldsymbol{i}}_k^c)$, where $\text{norm}(\boldsymbol{x}) = \boldsymbol{x}/\|\boldsymbol{x}\|$. We use $\boldsymbol{t}_q^c$ instead of $\tilde{\boldsymbol{t}}_q^c$ to introduce a residual connection (He et al., 2016) between $\boldsymbol{g}^c$ and the input text features, which enhances OOD detection and simplifies training (see Sec. 4.4 for experiments on $\lambda$). With this setup, the OOD Learner degenerates to MCM (Ming et al., 2022a) when $\lambda = 0$ or $K = 0$.

Our design is inspired by Q-Former (Li et al., 2023), as shown in Fig. 2 of their work. Unlike Q-Former, which extracts a single image feature and a single text feature from an image–text pair, our module is designed to extract a class feature from texts and images together.

### 3.3 LOSS FUNCTION

We adapt CLIP's contrastive loss (Radford et al., 2021; Khosla et al., 2020), originally designed for 1-to-1 text-image mapping, to our OOD training, which learns a 1-to-many mapping from a class representation to a distribution of images. The CLIP loss function:

$$\mathcal{L}^{\text{CLIP}} = \frac{1}{2}(\mathcal{L}^{i \to t} + \mathcal{L}^{t \to i}),$$

$$\mathcal{L}^{t \to i} = -\frac{1}{B}\sum_{m=1}^B \log \frac{\exp(\text{sim}(\boldsymbol{t}_m, \boldsymbol{i}_m)/\tau)}{\sum_{n=1}^B \exp(\text{sim}(\boldsymbol{t}_m, \boldsymbol{i}_n)/\tau)}, \quad \mathcal{L}^{i \to t} = -\frac{1}{B}\sum_{m=1}^B \log \frac{\exp(\text{sim}(\boldsymbol{i}_m, \boldsymbol{t}_m)/\tau)}{\sum_{n=1}^B \exp(\text{sim}(\boldsymbol{i}_m, \boldsymbol{t}_n)/\tau)},$$

where $(\boldsymbol{t}_m, \boldsymbol{i}_m)$ represents the features of a text-image pair extracted by the trainable CLIP model, $B$ is the batch size, and $\tau$ is the temperature parameter. For OOD detection, an image $\boldsymbol{i}^c$ is compared with class representations $\mathcal{G} = \{\boldsymbol{g}^1, ..., \boldsymbol{g}^C\}$, each representing a data distribution. The image $\mathbf{i}^c$ should have higher similarity to its own class representation $c$ and lower similarity to others. Thus, we design the following contrastive loss for pretraining $g(\cdot)$:

$$\mathcal{L}^g = \frac{1}{2}(\mathcal{L}^{i \to g} + \mathcal{L}^{g \to i}),$$

$$\mathcal{L}^{g \to i} = -\frac{1}{B}\sum_{c=1}^B \log \frac{\exp(\text{sim}(\boldsymbol{g}^c, \boldsymbol{i}^c)/\tau)}{\sum_{d=1}^B \exp(\text{sim}(\boldsymbol{g}^c, \boldsymbol{i}^d)/\tau)}, \quad \mathcal{L}^{i \to g} = -\frac{1}{B}\sum_{c=1}^B \log \frac{\exp(\text{sim}(\boldsymbol{i}^c, \boldsymbol{g}^c)/\tau)}{\sum_{d=1}^B \exp(\text{sim}(\boldsymbol{i}^c, \boldsymbol{g}^d)/\tau)}.$$

During pretraining, each batch contains $B$ classes of $Q$-shot texts $\mathcal{T}^c$, $K$-shot images $\mathcal{I}^c$ and an extra image $\boldsymbol{i}^c \notin \mathcal{I}^c$ but in the same class as $\mathcal{I}^c$. $\mathcal{T}^c$ and $\mathcal{I}^c$ are used to compute the class representation $\boldsymbol{g}^c$, while the extra image serves as an independent sample for loss computation. We highlight two key differences between the contrastive learning in CLIP and our approach:

- **Class representation source:** OOD Learner computes $\boldsymbol{g}^c$ from both images and texts rather than text-only, offering a richer and more accurate class representation.
- **Multiple positive candidates:** $\boldsymbol{i}^c$ could be any image in class $c$, so the class representation $\boldsymbol{g}^c$ includes multiple image candidates as potential positive samples in the loss. This approach enhances the alignment between class representation and their images, improving the robustness of the training process and the performance of OOD detection.

### 3.4 DATA CORPUS CURATION

Web-scale datasets like `CC3M` (Sharma et al., 2018b), `CC12M` (Changpinyo et al., 2021), and `LAION-400M` (Schuhmann et al., 2021) contain rich one-to-one text-image pairs. To pretrain OOD Learner, we curate the text-image pairs into one-to-many class-image sets, and treat each text as a pseudo-class to retrieve the most similar images. A naive approach is to rank the CLIP feature similarity between a text and all images, and select the top matches. However, for a dataset with $N$ text-image pairs, this requires $\mathcal{O}(N)$ time per text and $\mathcal{O}(N^2)$ for the full dataset, which is infeasible at a million-scale $N$.

To address this challenge, we use the method ANNOY (Indyk & Motwani, 1998), algorithm to approximate the most similar images for each pseudo-class. As shown in Fig. 4, we construct a binary tree from all image features. The same tree, once built, can be reused across all text features to find their approximate nearest images. Unlike the naive approach $\mathcal{O}(N^2)$, ANNOY reduces the complexity to $\mathcal{O}(N \log N)$ and makes large-scale retrieval feasible. We further detail our data curation process in Alg. 1, Appendix D.

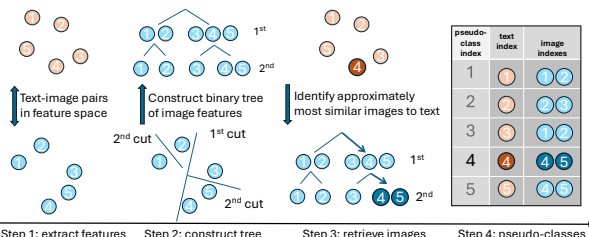

Figure 4: **A curation example with 5 text-image pairs and top-ranked number 2**. (1) Extract image and text features using CLIP. (2) Apply ANNOY (Indyk & Motwani, 1998) to construct a binary tree from image features. (3) Trace the tree to find approximately the most similar images to a given text. (4) Form a matrix of pseudo-classes.

**Implementation details.** For pretraining, we consider the following datasets[2]: CC3M, CC12M, and LAION-400M. Empirically, OOD Learner trained on LAION-400M achieves the best OOD detection performance. Due to computational constraints, we curate the OOD pretraining corpus from subset of LAION-400M dataset: LAION-0.4M, LAION-4M, and LAION-40M with ratio $^1/_{1000}$, $^1/_{100}$ and $^1/_{10}$ respectively. Multiple scales of LAION allow us to assess performance gains from scaling data. To report our performance, our experiments (Sec. 4) focus on the large-scale LAION-40M as the main dataset, which contains ∼28M valid text-image pairs.

Data curation is performed offline prior to pretraining. We selected the top 40 most similar images to account for different choices of $K$. The curation process introduces randomness, as shown in line 9 of Alg. 1. We repeat the curation 10 times with 10 binary decision trees constructed for better approximation. Computational cost for processing 28M text-image pairs is in Appendix F.

## 4 EXPERIMENTS

This section presents experiments and ablations designed to answer the following research questions:

**Q1** Can OOD Learner generalize to OOD detection tasks with different ID datasets? How does it compare to existing few-shot OOD detection methods?

**Q2** Can OOD Learner perform effectively with varying numbers ($k$) of few-shot samples without fine-tuning? How does the value of $k$ affect the performance?

**Q3** Can OOD Learner accommodate different numbers of ID classes, including one-class scenario?

**Q4** Are there additional important properties of OOD Learner, such as its scalability with data size?

Sec. 4.1 introduces benchmarks and baselines, then Sec. 4.2 and 4.3 evaluate OOD Learner and baselines on varied benchmarks to address **Q1**. Finally, Sec. 4.4 presents ablations for **Q2**, **3**, and **4**.

### 4.1 BENCHMARKS AND BASELINES

**Datasets.** Following the benchmark of CLIP-based OOD detection, we use ImageNet-1K (**I**) (Deng et al., 2009) as the ID dataset, with iNaturalist (**N**) (Van Horn et al., 2018), SUN (**S**) (Xiao et al., 2010), Places365 (**P**) (Zhou et al., 2017), and Texture (**T**) (Cimpoi et al., 2014) as OOD datasets. We further apply leave-one-out testing protocols among Places365, Texture, EuroSAT (**E**) (Helber et al., 2019), and Food101 (**F**) (Zhang et al., 2023) to create 4 additional ID→OOD settings. This allows us to evaluate OOD detection across diverse ID datasets.

**Baselines.** We consider the following baselines: MCM (Ming et al., 2022a), DCLIP (Menon & Vondrick, 2022), CLIPN (Wang et al., 2023), NegLabel (Jiang et al., 2024), CoOp (Zhou et al., 2022), and LoCoOp (Miyai et al., 2024). (Refer to Appendix E for the reason to choose these baselines.) Additionally, we introduce two CoOp variants: $\text{CoOp}^0$ and $\text{CoOp}^{\text{uni}}$. $\text{CoOp}^0$ removes data augmentation (*e.g.*, random cropping and flipping) from CoOp's few-shot training. (See Appendix E for the rationale of not modifying OOD Learner). $\text{CoOp}^{\text{uni}}$ trains CoOp with contrastive loss on pseudo-classes from LAION-40M to learn a universal prompt vector, denoted by the superscript "uni." We also append "-**I**" to CoOp trained on ImageNet and "-**L**" to CoOp trained on LAION-40M.

Table 1: **Comparison on few-shot OOD detection performance w/o fine-tuning (from ID→OOD).** OOD Learner consistently improves OOD detection performance across diverse ID datasets without requiring fine-tuning, whereas prompt-tuning-based methods fail to do so. We use bold abbreviations for dataset names: **I** for ImageNet-1K, **N** for iNaturalist, **S** for SUN, **P** for Places365, **T** for Texture, **E** for EuroSAT, and **F** for Food101.

| Method | I→NSPT | E→PTF | P→ETF | T→EPF | F→EPT | Average |
|---|---|---|---|---|---|---|
| #Classes in ID | 1000 | 10 | 365 | 47 | 101 | / |
| *ID Classification (Accuracy % ↑)* | | | | | | |
| CLIP (Radford et al., 2021) | 73.40 | 39.52 | 39.00 | 50.87 | 92.16 | / |
| *OOD Detection (AUROC % ↑)* | | | | | | |
| MCM (Ming et al., 2022a) | 91.65 | 67.97 | 81.18 | 85.08 | 99.55 | 85.09 |
| CLIPN (Wang et al., 2023) | 93.10 | **88.15** | 78.32 | 84.80 | 99.43 | 88.76 |
| NegLabel (Jiang et al., 2024) | **94.21** | 83.68 | 76.74 | 88.81 | **99.84** | 88.66 |
| CoOp-**I** (Zhou et al., 2022) | 93.63 | 69.24 | 77.22 | 79.47 | 99.51 | 81.81 |
| CoOp$^{uni}$-**L** | 88.91 | 66.15 | 77.48 | 90.61 | 99.35 | 84.50 |
| OOD Learner (ours) | 92.93 | 81.00 | **83.70** | **93.29** | 99.61 | **90.11** |

## 4.2 GENERALIZATION CAPABILITY

**Stronger generalization without finetuning.**  Table 1 presents that existing few-shot OOD methods struggle to generalize across ID datasets without fine-tuning, whereas our method generalizes robustly with no fine-tuning required. The different **ID→OOD** columns represent evaluations of OOD detection performance across various ID datasets. When CoOp-**I** performs well on **I→NSPT** (ImageNet-1K serves as the ID dataset) it performs poorly when other datasets serve as the ID dataset, *e.g.*, 77.22% on **P→ETF**, which is even worse than MCM (81.18%), suggesting that the tuned prompt vector on ImageNet-1K does not generalize to other ID datasets. We further train CoOp on the data corpus curated from LAION-40M in Sec. 3.4, namely CoOp$^{uni}$-**L**. We find that CoOp$^{uni}$-**L** still underperforms MCM on many ID datasets. In contrast, OOD Learner consistently outperforms MCM across all ID datasets, suggesting that OOD Learner effectively leverages context to better align ID class representations with ID images, and successfully learns the ID representation. In all **ID→OOD** cases, OOD Learner performs better than CoOp-**I** only except for **I→NSPT**, highlighting the strong generalization ability of OOD Learner compared to prompt tuning-based methods.

**The relationship between classification and OOD detection.**  Notably, as ID dataset changes from **F** to **T** to **P** to **E**, the classification task becomes progressively more challenging for CLIP, and the OOD detection performance of MCM also declines. We observe that OOD Learner provides a better performance boost over MCM as the classification task becomes harder. This suggests that when texts alone serve as biased representations of ID classes, OOD Learner effectively learns better ID class representations to achieve better alignment with ID images.

## 4.3 COMPARISON ON OOD BENCHMARKS

**Comparison with few-shot methods.**  CoOp and LoCoOp augment few-shot images with random flipping and cropping, "bootstrapping" tuned prompts for better ID alignment. In contrast, OOD Learner uses the few-shot images as-is. We therefore evaluate CoOp without augmentation, denoted as CoOp$^{0}$, and find that while OOD Learner outperforms CoOp$^{0}$ in OOD detection, it underperforms in ID classification, as shown in Table 2. This is because prompt tuning leverages classification loss on ID classes, anchoring tuned prompts more effectively for ID classification, whereas OOD Learner is trained solely to capture class representations, excelling in OOD detection. Overall, OOD Learner delivers significant OOD detection gains without fine-tuning, achieving competitive results.

**Comparison with zero-shot methods.**  We highlight MCM as the primary zero-shot baseline that OOD Learner builds upon. Our results demonstrate that OOD Learner improves both classification and OOD detection performance using few-shot examples. As shown in Table 1, zero-shot methods such as NegLabels achieve stronger performance 94.21% AUROC on **I→NSPT**. A followup work, AdaNeg (Zhang & Zhang, 2024), further improves this score to 96.66%. Although these zero-shot methods outperform OOD Learner on **I→NSPT**, our work addresses a different setting, few-shot learning, and approaches on the two settings could be combined. We do not pursue this direction, as it is incremental. Instead, we aim to introduce the novel concept of OOD Learner and provide the first proof-of-concept that OOD Learner can enhance OOD detection performance with few-shot examples, without requiring fine-tuning.

Table 2: **OOD detection performance comparison with ImageNet-1K (Deng et al., 2009) as ID.** (CLIP-L/14 as feature encoders) While existing methods require prompt tuning on a specific ID dataset to function, OOD Learner is finetuning-free and achieves competitive performance.

| Method | Finetuning Requirement | Few-Shot Image Augmentation | Classification (Acc% ↑) | OOD Detection (AUROC% ↑) | | | | |
|---|---|---|---|---|---|---|---|---|
| | | | | iNaturalist | SUN | Places | Texture | Average |
| MCM | N/A | N/A | 73.40 | 95.11 | 94.30 | 92.16 | 85.04 | 91.65 |
| DCLIP | N/A | N/A | 75.62 | 94.55 | 94.05 | 92.37 | 82.95 | 90.98 |
| LoCoOp | Yes | Yes | 78.30 | 96.30 | 95.50 | 93.07 | 90.47 | 93.84 |
| CoOp | Yes | Yes | 78.15 | 96.09 | 95.29 | 92.85 | 90.26 | 93.63 |
| CoOp$^0$ | Yes | No | 77.43 | 95.30 | 95.08 | 93.33 | 87.26 | 92.74 |
| CoOp$^{\text{uni}}$-L | No | No | 69.22 | 90.70 | 90.20 | 87.85 | 86.90 | 88.91 |
| OOD Learner | No | No | 76.66 | 96.46 | 94.42 | 92.59 | 88.24 | 92.93 |

**Computational efficiency.** Few-shot OOD detection involves two stages: (1) few-shot training and (2) OOD detection. In stage (1), our method is more efficient than traditional few-shot approaches, which require fine-tuning for each ID dataset, whereas we only need inference; zero-shot methods like MCM incur no cost at all. In stage (2), costs are nearly identical across methods: CLIP processes test images and compares them with precomputed class representations; methods with negative labels (e.g., NegLabel (Jiang et al., 2024)) introduce additional inner products for negative labels, but this overhead is negligible relative to CLIP inference of images.

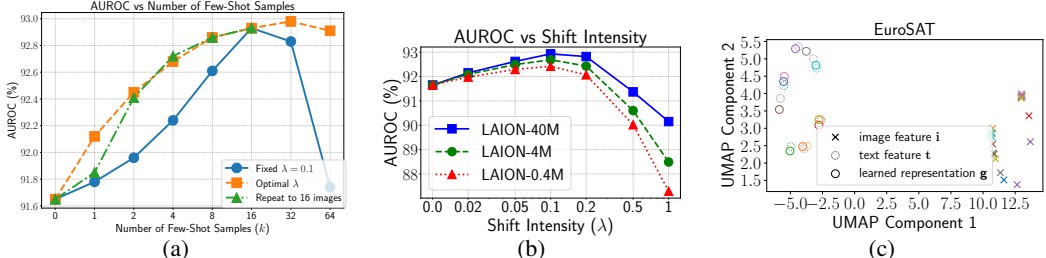

(a)            (b)            (c)

Figure 5: **(a) OOD Learner works with varying number of few-shot samples.** OOD Learner is trained with a fixed number of in-context samples ($k = 16$) yet demonstrates length generalization across a range of sample sizes, meaning it can handle varying numbers of few-shot samples without requiring any fine-tuning. **(b) Effect of shift intensity on OOD detection performance.** The three curves represent OOD Learners trained on different data scales, sharing a sweet point at $\lambda = 0.1$. **(c) UMAP (McInnes et al., 2018) visualization of learned ID class representations.** OOD Learner learns ID class representation $\boldsymbol{g}$, aiming to achieve better alignment with the image distribution.

### 4.4 ABLATION STUDIES

**Flexibility on few-shot samples.** Fig. 5a shows how OOD Learner performs with varying numbers of few-shot samples. To improve readability, we restate the formulation of the learned ID class representation from Sec. 3.1 for number of few-shot texts equal to 1: $\boldsymbol{g}^c = \text{norm}(\boldsymbol{t}_1^c + \lambda \cdot \sum_{k=1}^{K} \tilde{\boldsymbol{i}}_k^c)$. Here, $\boldsymbol{t}_1^c$ represents the CLIP text feature, which is regarded as the ID class representation for zero-shot OOD detection by MCM, and $\tilde{\boldsymbol{i}}_k^c$ represents the output of attention module corresponding to each image. During training, we set $\lambda = 1$, and for inference, we use $\lambda = 0.1$.

For OOD detection with different numbers of few-shot samples, we explore three approaches below:

- **Fixed $\lambda = 0.1$:** We consistently set $\lambda = 0.1$ and use the few-shot images in their original form.
- **Optimal $\lambda$:** We dynamically select the optimal $\lambda$ from $\{1.0, 0.5, 0.2, 0.1, 0.05, 0.02, 0.01\}$ for different number of few-shot samples, maximizing OOD detection performance. While effective, this setting assumes some knowledge of optimal values.
- **Repeat to 16 images:** When fewer than 16 images are available, we repeat images to a total of 16.

Fig. 5a demonstrates that OOD Learner can flexibly handle varying few-shot sample counts. OOD Learner is pretrained with 16 few-shot samples, and its performance begins to decline when $k$ reaches 32 or 64. This is due to the generalization issue on the number of input length. We provide more information on this in Appendix E.4.

**Optimal value of shift intensity $\lambda$.** Fig. 5b illustrates the performance change on the few-shot OOD detection benchmark as the shift intensity $\lambda$ increases. The three bell curves share a common

Figure 6: **OOD Learner detects OOD samples with a varied number of ID classes.** The dashed vertical lines are dynamic thresholds to split test images into the ID set and OOD set. The red small cross mark with a white background indicates a mistake by OOD Learner.

peak at $\lambda = 0.1$, indicating an optimal value. Thus, we empirically determine $\lambda$ and use $0.1$ for all experiments.Notably, $\lambda$ is analogous to key hyperparameters in related methods, such as the prefix length in COOP and the top-$k$ similar image regions to a text qeurt in LOCOOP, all of which require empirical tuning. In Fig. 5b, we conduct a similar ablation study by varying this key parameter, as done in COOP and LOCOOP. We further illustrate how the OOD Learner positions the class representation from $t$ ($\lambda = 0$) to $g$ ($\lambda = 0.1$) on the dataset EuroSAT in Fig. 5c.

**Modality Gap.** Fig. 5c shows a significant modality gap between image and text features, consistent with findings by Liang et al. (2022) across various pretrained VLMs. OOD Learner learns ID class representations using few-shot images while preserving this modality gap. Notably, fully bridging the modality gap has been found harmful to classification (Liang et al., 2022).

**Effect of data and model scaling.** Table 3 report the performance of OOD Learner pretrained on LAION-0.4M, LAION-4M, and LAION-40M, demonstrating that its performance scales with data and model size. Fig. 7 in Ap-

Table 3: **Data and model scaling impact on AUROC:** Scaling up data and model size improves the AUROC score on ImageNet-1K from 92.42 to 92.93, indicating enhanced performance.

| Dataset | #samples | #layers | #hidden dimension | AUROC% ↑ |
|---|---|---|---|---|
| LAION-0.4M | 281,877 | 2 | 384 | 92.42 |
| LAION-4M | 2,863,904 | 2 | 768 | 92.69 |
| LAION-40M | 28,611,183 | 2 | 1,536 | 92.93 |

pendix E.3 further shows that **noise in pseudo-classes** correlates with the size of the raw image-text data, and Table 5 in Appendix F shows the computational cost over different data sizes.

**Versatility on ID classes.** Fig. 6 demonstrates the versatility of OOD Learner on ID classes, using 16 images from 8 ImageNet-1K classes for testing, and each class is paired with a visually similar class. OOD Learner is pretrained and tested with only few-shot images (excluding text, since CLIP alone already achieves high ImageNet-1K accuracy). It performs well on both single- and multi-class datasets, as each class representation relies solely on its own context. In contrast, prompt tuning-based methods require multi-class training, making it infeasible for single-class datasets and yielding non-invariant, entangled class representations.

## 5 CONCLUSION

In this paper, we introduce "OOD Learner," a versatile framework for few-shot OOD detection based on the paradigm of in-context learning. To enable effective pretraining, we develop a data curation strategy that transforms web-scale text-image datasets into structured pretraining corpora with large-scale pseudo-classes. Unlike CLIP-based methods that require separate prompt tuning per ID dataset, our OOD Learner generalizes across diverse ID distributions without fine-tuning, achieving SOTA performance and efficiency in few-shot OOD detection. We hope this work paves the way for more adaptive, scalable few-shot OOD detection with VLMs.

## REPRODUCIBILITY STATEMENT

All results in this paper are reproducible. For our main experiments (*e.g.*, OOD detection on ImageNet-1K as ID dataset), we evaluate OOD Learner with 50 runs, achieving a mean AUROC of 92.93. Few-shot baselines are tested with three runs: CoOp (mean 93.63; raw 93.69/93.42/93.78) and LoCoOp (mean 93.83; raw 93.57/94.14/93.80). AUROC values for OOD Learner across 50 runs range from 0.9278 to 0.9304. Codes will be released upon acceptance.

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

## A  THE USE OF LARGE LANGUAGE MODELS (LLMS)

LLMs are only used in this paper to improve writing. The author drafts the text, then iterates with LLMs on one to several sentences at a time to improve readability. The author is responsible for the check of the final text, to make sure the information is delivered precise and accurate.

## B  LIMITATIONS AND FUTURE WORKS

We acknowledge the following limitations in our work:

- **Pretraining Data Quality and Quantity**: The pretraining data could be further improved. While we have used a subset of `LAION-400M` Schuhmann et al. (2021), future efforts could involve leveraging advanced vision-language models (VLMs) to verify whether the retrieved images correspond accurately to their pseudo-classes, thereby improving data quality. Additionally, using the full `LAION-400M` dataset, or even the `LAION-5B` Schuhmann et al. (2022) dataset, could significantly enhance the quantity of training data.
- **Text Versatility in OOD Detection**: The role of text in OOD detection has not been fully explored in this study. Although preliminary experiments with multiple text inputs—similar to those in DLIP—were conducted, we found no substantial improvement in OOD detection using `ImageNet-1K` as the ID dataset. This aspect requires further exploration, particularly when pretraining data of higher quality is available.

Future research will focus on addressing limitations and expanding on key aspects of current work:

- **Enhancing Pretraining Data**: To improve both the quality and quantity of pretraining data, we plan to explore the full `LAION-400M` or even the `LAION-5B` dataset. We will also aim to balance class distributions and filter out false in-class images using advanced VLMs to refine the data further.
- **Exploring Text Inputs for OOD Detection**: Further investigation is needed to explore the use of multiple text inputs for OOD detection. This could potentially reveal new ways to improve performance, especially with higher-quality pretraining data.
- **Generalizing Across Datasets**: We also plan to evaluate the performance of our OOD detection system on a wider range of in-distribution datasets to better assess the generalizability of our approach.

## C  NOTATIONS

We summarize all notations used by the paper in Table 4 to improve readability.

## D  PSEUDO ALGORITHM

We share pseudo algorithm for our data curation process in Alg 1.

## E  CLARIFICATION FOR POTENTIAL QUESTIONS

This section explains the logic behind our implementation, addressing questions that may arise as readers go through our paper.

### E.1  REASON TO CHOOSE BASELINES: WHAT ABOUT OTHER CLIP-BASED OOD DETECTION METHOD?

Our method can enhance the performance of zero-shot approaches: to validate our "finetuning-free few-shot" setting, we set MCM as a baseline and demonstrate that our OOD learner boosts MCM's performance from 91.65 to 92.93. For few-shot baselines ID-Like (Bai et al., 2024), NegPrompt (Li et al., 2024), and GalLoP (Lafon et al., 2024), we encounter issue and are not able to reproduce

Table 4: **Notations.** We summarize all notations used by the paper to improve readability.

| Notation | First-time appearance | Explanation |
|---|---|---|
| C | Sec. 3.1 | number of ID classes |
| $\mathcal{C}$ | Sec. 3.1 | a set of ID classes |
| $c^i$ | Sec. 3.1 | the $i^{\text{th}}$ ID class |
| i | Sec. 3.1 | image |
| t | Sec. 3.1 | test |
| $i$ | Sec. 3.1 | image feature |
| $t$ | Sec. 3.1 | test feature |
| $\mathcal{I}^c$ | Sec. 3.1 | a set of ID images in class $c$ |
| $\mathcal{T}^c$ | Sec. 3.1 | a set of ID texts in class $c$ |
| $K$ | Sec. 3.1 | number of few-shot images |
| $Q$ | Sec. 3.1 | number of few-shot texts |
| $g$ | Sec. 3.1 | OOD Learner |
| $g$ | Sec. 3.1 | the learned class representation by OOD Learner |
| sim | Sec. 3.1 | similarity function, by default cosine similarity |
| $L$ | Sec. 3.2 | number of attention layers |
| $q$ | Sec. 3.2 | query of attention |
| $k$ | Sec. 3.2 | key of attention |
| $v$ | Sec. 3.2 | value of attention |
| $x$ | Sec. 3.2 | input feature of attention |
| $y$ | Sec. 3.2 | output feature of attention |
| $h$ | Sec. 3.2 | feature output by weighted attention |
| $\tilde{t}$ | Sec. 3.2 | processed feature after $L$ layers of self- and cross-attention |
| $\tilde{i}$ | Sec. 3.2 | processed feature after $L$ layers of self- and cross-attention |
| $\lambda$ | Sec. 3.2 | a hyperparameter to deal with generalization issue |
| $d$ | Sec. 3.2 | dimension of features |
| $B$ | Sec. 3.3 | batch size |
| $\gamma$ | Sec. 3.3 | temperature of CLIP |
| $\mathcal{G}$ | Sec. 3.3 | the set of representations of ID classes, $\mathcal{G} = \{g^1, \ldots, g^C\}$ |
| $N$ | Sec. 3.4 | the number of text-image pairs in the raw dataset |
| $R$ | Sec. 3.4 | the number of the top-$R$ approximately similar images to text $\mathbf{t}_n$ |
| $M$ | Sec. 3.4 | a $N \times R$ matrix with $n^{\text{th}}$ row containing the top-$R$ approximately similar images to text $\mathbf{t}_n$ |
| $\mathcal{V}$ | Sec. 3.4 | the set of images corresponding to a leaf node in the tree structure |
| $\mathcal{Q}$ | Sec. 3.4 | the queue containing nodes in a tree |
| $\mathcal{N}$ | Sec. 3.4 | the set of leaf nodes in a tree |
| $\mathcal{O}$ | Sec. 3.4 | complexity of algorithm |

---

**Algorithm 1** PSEUDO-CLASS CREATION VIA ANNOY

---

**Input:** Text-image pairs $\{\mathbf{t}_n, \mathbf{i}_n\}_{n=1}^N$, pretrained CLIP model, maximum leaf node size $l$, top-ranked number $R$.

**Output:** $\boldsymbol{M} \in \mathbb{R}^{N \times R}$ (each row represents a pseudo-class).

1: **Compute CLIP features:** $\{\mathbf{t}_n, \mathbf{i}_n\}_{n=1}^N \xrightarrow{\text{CLIP}} \{\boldsymbol{t}_n, \boldsymbol{i}_n\}_{n=1}^N$.
2: **Initialize matrix:** $\boldsymbol{M} \leftarrow \boldsymbol{0}_{N \times R}$.
3: **Construct a binary tree using $\{\boldsymbol{i}_n\}_{n=1}^N$:**
4: Initialize root node as $\mathcal{V} = \{\boldsymbol{i}_n\}_{n=1}^N$.
5: Initialize the queue $\mathcal{Q} = \{\mathcal{V}\}$, the set of leaf nodes $\mathcal{N} = \{\}$.
6: **while** $\mathcal{Q}$ is not empty:
7:     Pop a node from queue $\mathcal{Q} \rightarrow \mathcal{V}'$.
8:     **if** $|\mathcal{V}'| > l$:
9:         Select two random points $\boldsymbol{i}_A, \boldsymbol{i}_B \in \mathcal{V}'$.
10:       Get mean $\boldsymbol{m} = \frac{\boldsymbol{i}_A + \boldsymbol{i}_B}{2}$ and difference $\boldsymbol{d} = \boldsymbol{i}_A - \boldsymbol{i}_B$.
11:       Split $\mathcal{V}'$ into left and right child nodes:

$$\text{left}(\mathcal{V}') = \mathcal{V}_0 = \{\boldsymbol{x} \in \mathcal{V}' : \langle \boldsymbol{x} - \boldsymbol{m}, \boldsymbol{d} \rangle \geq 0\}, \tag{7}$$

$$\text{right}(\mathcal{V}') = \mathcal{V}_1 = \{\boldsymbol{x} \in \mathcal{V}' : \langle \boldsymbol{x} - \boldsymbol{m}, \boldsymbol{d} \rangle < 0\}. \tag{8}$$

12:       Add $\mathcal{V}_0$ and $\mathcal{V}_1$ to the queue $\mathcal{Q}$.
13:     **else**:
14:       Add $\mathcal{V}'$ to the set of leaf nodes $\mathcal{N}$.
15: **Approximate nearest neighbor retrieval with $\{\boldsymbol{t}_n\}_{n=1}^N$:**
16: **for** $n = 1$ to $N$ **do**
17:     Route $\boldsymbol{t}_n$ through the tree based on Eqn. 7-8 to the leaf node $\mathcal{V}(\boldsymbol{t}_n) \in \mathcal{N}$.      ▷ Time complexity: $\mathcal{O}(\log N)$
18:     Compute $\text{sim}(\boldsymbol{t}_n, \boldsymbol{i})$ for all $\boldsymbol{i} \in \mathcal{V}(\boldsymbol{t}_n)$.
19:     Select image indices corresponding to the top-$R$ highest similarities and update: $\boldsymbol{M}_{n,:} \leftarrow [\text{index}^{(1)}, \dots, \text{index}^{(R)}]$.
20: **end for**

---

their results as also indicated by many other researchers in their GitHub repo, which makes we can only copy their paper's results rather reproduce their results and then examine their methods on our evaluation as in the main paper Table 1. CLIPScope (Fu et al., 2024) and DualAdaptor (Chen et al., 2024) are unpublished arXiv works and we simply include them in Related Work.

## E.2 WHY NOT APPLY DATA AUGMENTATION ON OOD LEARNER?

The attention mechanism has $O(L^2)$ complexity and $L$ is the number of image samples to the attention mechanism. COOP is trained with 100 epochs and randomly augments the images one time per epoch. For a fair comparison, we need to augment 16-shot to 1600-shot, which leads to an infeasible $10000\times$ computation cost. A better way to compute attention across large $L$ could be a future research direction.

## E.3 HOW DOES THE SIZE OF RAW TEXT-IMAGE DATA AFFECT THE QUALITY OF PSEUDO-CLASSES?

Due to the pseudo-classe creation process, the quality of the processed pseudo-classes is correlated with the size of the pretraining data. We share such correlation in Fig. 7, showing that larger size of pretraining data leads to higher quality (higher inner consistency, *e.g.*, the images in a pseudo-class are more likely to follow the same concept.) of processed pseudo-classes.

## E.4 WHY THE PERFORMANCE INCREASE AND DECREASE IN MAIN PAPER FIG. 5A?

Our model is trained on 16-shot images. When meeting beyond 16-shot images, one can consider this to be an OOD case (*e.g.*, the input with 32 images is out of the distribution of input with 16 images), hence the performance drops. Similar phenomenon has been known in the literature of Large Language Models which is also based on attention mechanism. For instance, Modarressi et al. (2025)

Figure 7: **Larger size $\Rightarrow$ higher quality**.

point out large language models perform well in short contexts, performance degrades significantly as context length increases.

## F WHAT IS THE COMPUTATION/MEMORY COST ON PROCESSING 28M TEXT-IMAGE PAIRS?

28M images are converted to CLIP features with size 28M$\times$768 (float16, 43GB). The pseudo-classes, *i.e.*, the text-to-image mapping is converted to a tensor with size 28M$\times$40 (int32, 4.5GB, top 40 similar images' indexes are saved, top 17 used in experiments). Loading these large tensors takes 20-30 mins. We provide following Table 5 on time cost on processing and training with these tensors. We consider future work on how to find the golden classes among so many pseudo-classes.

| #Raw Images | Extract CLIP Feature (4xA100) | Curate Pseudo-Classes (GCP, 128 core CPU) | Pretraining (1xA100) |
|---|---|---|---|
| $\sim$30M | $\sim$2 days | $\sim$2 days | $\sim$6 hours (1 epochs) |
| $\sim$3M | $\sim$4 hours | $\sim$2 hours | $\sim$2 hours (4 epochs) |
| $\sim$0.3M | $\sim$30 mins | $\sim$15 mins | $\sim$3 hours (20 epochs) |

Table 5: Computational cost on varied data sizes.

## G EXPERIMENTAL SETUP

We share the details of pretraining hardware and hyperparameters in this section.

We use 4$\times$NVIDIA A100 Tensor Core GPU (40GB version) for our experiments. The OOD learner ends up pretraining with a small batch size of 32 on 1$\times$NVIDIA A100 Tensor Core GPU (40GB version), while larger batch sizes on 4$\times$NVIDIA A100 Tensor Core GPU (40GB version) in parallel

lead to worse performance. Although one GPU is sufficient, the pretraining process is memory-intensive due to the need to load image and text feature data. For LAION-40M, each feature tensor, with a size of $\sim 30M \times 768$ in float16, consumes approximately 40GB of memory.

For hyperparameters of OOD Learner, we share the details in the following tables.

Table 6: Pretraining hyperparameters of OOD learner.

| Set Up | Hyperparameter | Value |
|---|---|---|
| OOD Learner | #hidden dimensions | 1,536 |
| | #layers | 2 |
| | #attention heads | 2 |
| Loss Function | temperature | 0.07 |
| Optimization | optimizer | AdamW Loshchilov (2017) |
| | learning rate | 1e-5 |
| | weight decay | 1e-5 |
| | #epochs | 1 |
| Dataloader | batch size | 32 |
| | #texts per class (Q) | 1 |
| | #images per class (K) | 16 |

## H DETAILED BREAKDOWN OF GENERALIZATION

In this section, we provide more details of Table 1, *i.e.*, instead of providing an aggregation performance, we provide a detailed breakdown of the performance for individual ID-OOD pair in Tables 7, 8, 9 and 10.

Table 7: **E→PTF**.

| Method | ID→OOD (AUROC % ↑) | | | |
|---|---|---|---|---|
| | **E→P** | **E→T** | **I→F** | Average |
| MCM | 47.62 | 70.70 | 85.61 | 67.97 |
| CoOp-**I** | 55.09 | 74.58 | 78.06 | 69.24 |
| CoOp$^{uni}$-**L** | 46.21 | 71.00 | 81.26 | 66.15 |
| OOD Learner (ours) | **65.24** | **84.33** | **93.44** | **81.00** |

Table 8: **P→ETF**.

| Method | ID→OOD (AUROC % ↑) | | | |
|---|---|---|---|---|
| | **P→E** | **P→T** | **P→F** | Average |
| MCM | 79.96 | 87.57 | **76.03** | 81.18 |
| CoOp-**I** | 88.81 | 88.88 | 53.97 | 77.22 |
| CoOp$^{uni}$-**L** | 81.86 | 81.61 | 68.97 | 77.48 |
| OOD Learner (ours) | **87.54** | **90.34** | 73.22 | **83.70** |

Table 9: **T→EPF**.

| Method | ID→OOD (AUROC % ↑) | | | |
|---|---|---|---|---|
| | **T→E** | **T→P** | **T→F** | Average |
| MCM | 85.82 | 84.65 | 84.83 | 85.08 |
| CoOp-**I** | 89.60 | 84.80 | 64.00 | 79.47 |
| CoOp$^{uni}$-**L** | **96.41** | 89.66 | 85.77 | 90.61 |
| OOD Learner (ours) | 96.30 | **93.02** | **92.54** | **93.29** |

Table 10: **F→EPT**.

| Method | ID→OOD (AUROC % ↑) | | | |
| --- | --- | --- | --- | --- |
| | **F→E** | **F→P** | **F→T** | Average |
| MCM | 99.98 | 99.63 | 99.05 | 99.55 |
| CoOp-**I** | 99.98 | 99.64 | 98.92 | 99.51 |
| CoOp$^{uni}$-**L** | 99.98 | 99.45 | 98.62 | 99.35 |
| OOD Learner (ours) | **99.99** | **99.71** | **99.13** | **99.61** |

# I  UMAP VISUALIZATION ON LEARNED ID CLASS REPRESENTATIONS

In addition to Fig. 7(b) presenting the UMAP visualization on `EuroSAT` (**E**), we provide additional visualizions on `ImageNet-1K` (**I**), subset of `ImageNet-1K` (subset-**I**), `Places365` (**P**), `Texture` (**T**), and `Food101` (**F**). Due to the large number of classes, we visualize only 20 randomly selected classes from the available set for each dataset, while fitting UMAP using all classes. One can observe a marginal distance between $t$ and $g$ in the visualization of ImageNet-1K (Fig. 8), while larger distances are evident in the other figures (Fig. 9, Fig. 10, and Fig. 11).

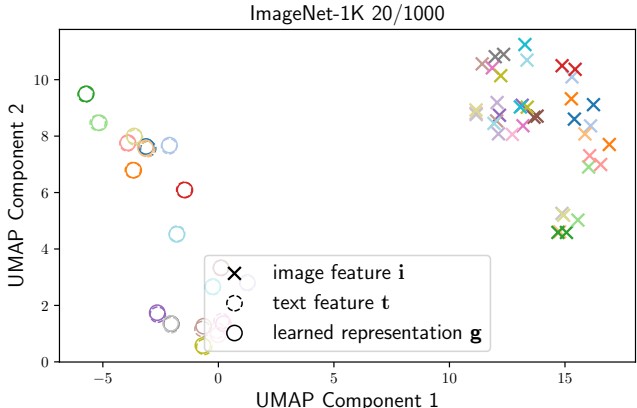

Figure 8: UMAP visualization on 20 randomly sampled classes of ImageNet-1K.

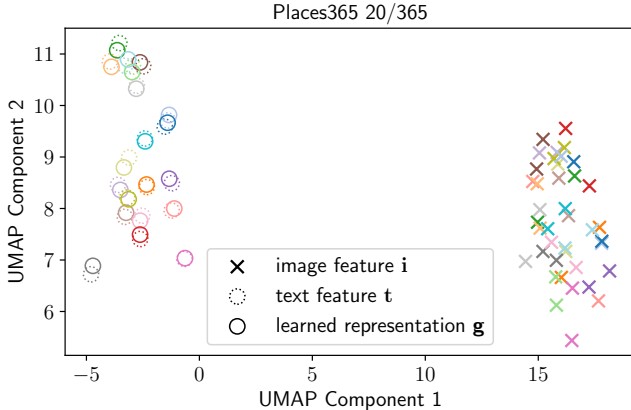

Figure 9: UMAP visualization on 20 randomly sampled classes of Places365.

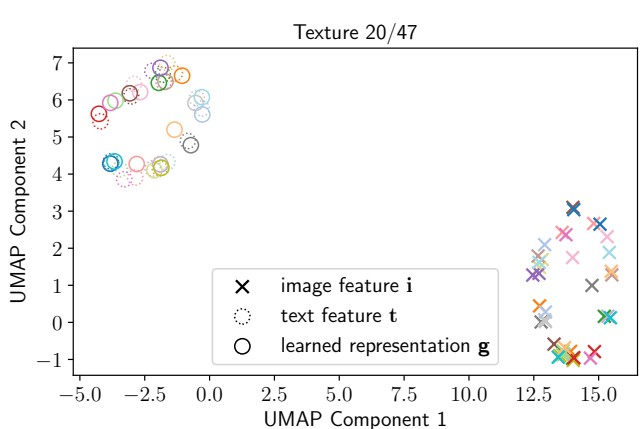

Figure 10: UMAP visualization on 20 randomly sampled classes of Texture.

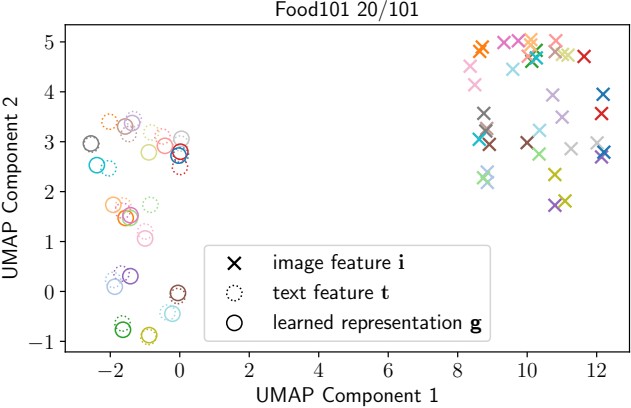

Figure 11: UMAP visualization on 20 randomly sampled classes of Food101.

