# OpenReview forum: "OOD Learner via In-Context Learning"
_ICLR.cc/2026/Conference — Submitted to ICLR 2026_

### Official Review · Reviewer_UskH · 2025-10-27

**Soundness:** 3
**Presentation:** 3
**Contribution:** 2
**Rating:** 4
**Confidence:** 4

**Summary:**

### Background
- This paper focuses on the few-shot OOD detection task.
- Prior works usually use the few-shot ID samples to refine text prompts. The authors argue that prompt tuning has several limitations.

### Method
- Therefore, the authors propose a universal OOD learner.
    - The OOD learner takes text and images from an ID class as input and outputs the class representation.
    - It is an attention-based module and trained with contrastive loss.
- The authors apply the ANNOY algorithm to process a large-scale dataset and pretrain the OOD learner.

### Results
- The proposed OOD learner achieves SoTA performance and efficiency in few-shot OOD detection.

**Strengths:**

- The paper is well-written with clear figures and tables.
- The method is clear and easy to understand.
- The experiments are comprehensive.

**Weaknesses:**

### Major: Do we really need large-scale training for OOD detection?
- (Gain vs cost) As I recall, MCM and NegLabel do not need extra training or ID examples, and they already get impressive accuracy. Although the proposed method outperforms MCM and NegLabel, the improvement is relatively modest (90.11 vs 88.66 in Table 1, and no comparison with NegLabel on ImageNet-1K). Considering the OOD learner needs to be trained on about 28M text-image pairs, I'm not sure if the performance gain is worth the computational cost.
- (Scaling) The scaling-up property of the OOD learner is not clear. As shown in Table 3, when the number of samples increases from 281K to 28M (100 times), the AUROC goes from 92.42 to 92.93. The gain is very small and maybe not statistically significant.
- (Test-time adaptation) Besides, recent test-time adaptation methods such as AdaND and AdaNeg show improved performance in OOD detection. How is the OOD learner compared with these methods? Could we implement AdaNeg in the few-shot OOD detection setting?


### Minor
- How does the method compare with NegLabel on ImageNet-1K?
- (Line 166) "without relyingper-dataset fine-tuning" missing space.

**Questions:**

Please see the Weaknesses section.

---

> ### Author Response · Authors · 2025-12-02
>
> [1] (Gain vs cost) As I recall, MCM and NegLabel do not need extra training or ID examples, and they already get impressive accuracy. Although the proposed method outperforms MCM and NegLabel, the improvement is relatively modest (90.11 vs 88.66 in Table 1, and no comparison with NegLabel on ImageNet-1K). Considering the OOD learner needs to be trained on about 28M text-image pairs, I'm not sure if the performance gain is worth the computational cost.
>
> The reviewer is right to doubt whether the trade-off is worth or not. But when considering that this is the first work to show non-finetuning few-shot OOD detection method can work, the answer is yes! It's similar to diffusion, before it beats GAN, any research towards beating GAN is meaningful.
>
> [2] (Scaling) The scaling-up property of the OOD learner is not clear. As shown in Table 3, when the number of samples increases from 281K to 28M (100 times), the AUROC goes from 92.42 to 92.93. The gain is very small and maybe not statistically significant.
>
> The authors acknowledge this weakness. The underlying reason of this phenomenon is currently unclear. It's worth to explore to increase the data effeciency further.
>
> [3] (Test-time adaptation) Besides, recent test-time adaptation methods such as AdaND and AdaNeg show improved performance in OOD detection. How is the OOD learner compared with these methods? Could we implement AdaNeg in the few-shot OOD detection setting?
>
> We regard AdaNeg and AdaND as zero-shot OOD detection methods. OOD learner is the first work on finetuning-free few-shot OOD detection method. We ackowledge that the performance gap compared with zero-shot methods. Zero-shot and few-shot methods are orthogonal and we believe it's possible to apply any zero-shot method on few-shot methods. We avoid to do since research-wise the contribution is too incremental.
>
> [4] How does the method compare with NegLabel on ImageNet-1K?
>
> It's shown in Table 1 under column I→NSPT.
>
> [5] (Line 166) "without relyingper-dataset fine-tuning" missing space.
>
> Thank you for the suggestion, will revise.

---

### Official Review · Reviewer_727E · 2025-10-27

**Soundness:** 3
**Presentation:** 3
**Contribution:** 3
**Rating:** 6
**Confidence:** 4

**Summary:**

Existing OOD detection frameworks typically follow the classic experimental setting, which requires training or fine-tuning on in-distribution (ID) datasets. This work departs from that paradigm by introducing a novel OOD detection framework capable of adapting to arbitrary ID datasets using only a few ID samples, without any additional training or fine-tuning. To achieve this, the authors propose an attention module that jointly learns class representations from both text and corresponding images. Moreover, to efficiently construct one-to-many class–image training sets, the framework employs ANNOY to retrieve the most similar images for each class with reasonable computational cost. In terms of performance, the proposed method achieves state-of-the-art AUROC results.

**Strengths:**

- The paper is well-written and easy to follow.
- The proposed architecture and data curation pipeline for pre-training are sensible.
- The experiments are extensive.

**Weaknesses:**

Weaknesses:

- I understand that the OOD learner is inspired by the in-context learning phenomenon observed in large language models. However, the authors should better motivate and clarify why the proposed architecture, in particular, the attention module,  is expected to work and provide justification for the specific design choices made.
-  The OOD evaluation does not fully follow the standard OOD benchmarking protocol, see https://zjysteven.github.io/OpenOOD/.
   - Specifically, the results on OpenImage-O are missing when ImageNet-1k is used as the ID dataset.
-  Experiment-wise, the authors should also report FPR95 values to provide a comprehensive evaluation of the proposed framework.


Minor weaknesses:
- While the paper includes extensive experiments on far-OOD detection benchmarks, it would be valuable to also include results on near-OOD benchmarks (\eg, ImageNet-1k as ID dataset). This would help determine whether the proposed design maintains strong performance under near-OOD conditions.
-   Some related works are missing. For CLIP-based zero-shot OOD detection, GL-MCM [1] and TAG [2] do not require OOD labels and fine-tuning.

 [1] GL-MCM: Global and Local Maximum Concept Matching for Zero-Shot Out-of-Distribution Detection, IJCV, 2023.

 [2] TAG: Text Prompt Augmentation for Zero-Shot Out-of-Distribution Detection, ECCV, 2024.

**Questions:**

See weaknesses.

---

> ### Author Response · Authors · 2025-12-02
>
> [1] I understand that the OOD learner is inspired by the in-context learning phenomenon observed in large language models. However, the authors should better motivate and clarify why the proposed architecture, in particular, the attention module, is expected to work and provide justification for the specific design choices made.
>
> We have tried different architectures. Motivated by Q-former in "BLIP-2: Bootstrapping Language-Image Pre-training with Frozen Image Encoders and Large Language Models", we designed the current attention module and found it works out.
>
> [2] The OOD evaluation does not fully follow the standard OOD benchmarking protocol, see https://zjysteven.github.io/OpenOOD/
>
> Could the author provides more information? We follow existing work such as MCM and CoOp, using the same training and testing datasets and image/text processing as them. Any reason why ours is not standard? We will surely consider the protocol shared by the reviewer.
>
> [3] Experiment-wise, the authors should also report FPR95 values to provide a comprehensive evaluation of the proposed framework.
>
> We appreciate the reviewer for this suggestion and will add those information to the appendix due to page limitation.
>
> [4] Some related works are missing. For CLIP-based zero-shot OOD detection, GL-MCM [1] and TAG [2] do not require OOD labels and fine-tuning.
>
> We appreciate the information and will add those work to the section of related work.

---

### Official Review · Reviewer_gejZ · 2025-10-28

**Soundness:** 3
**Presentation:** 3
**Contribution:** 2
**Rating:** 4
**Confidence:** 4

**Summary:**

This work focuses on few-shot out-of-distribution (OOD) detection and proposes a general In-Context Learning (ICL) framework, OOD Learner. To address the limitations of CLIP-based prompt-tuning methods—which require per–in-distribution (ID) dataset fine-tuning and exhibit limited generalization—the authors adopt a text–image dual-branch attention module. Using a small number of text–image pairs as in-context examples, the method adapts to arbitrary ID datasets without fine-tuning. During pretraining, the approach leverages large-scale text–image corpora (e.g., LAION-400M) and uses ANNOY to construct a massive set of pseudo-classes, thereby injecting generic OOD detection capability. Experiments on multiple ID datasets claim state-of-the-art (SOTA) performance and efficiency for OOD detection, with flexible adaptation to varying numbers of shots and to single-class scenarios, demonstrating strong generalization and practical utility.

**Strengths:**

1. By applying ANNOY to LAION-400M and related large-scale corpora, the method constructs roughly 30 million pseudo-classes, offering semantic coverage far beyond ImageNet-1K (1K classes).

2. Compared with prompt-tuning paradigms, the framework accommodates a variable number of shots without modifying the model, and it supports diverse ID settings including single-class and multi-class scenarios.

**Weaknesses:**

1. Pseudo-class construction relies on CLIP feature retrieval, which introduces potential mismatch risk.

2. Under the setting where ImageNet-1K serves as the ID dataset, the OOD detection performance does not show significant improvement.

3. The OOD Learner requires pretraining, leading to substantial computational cost; relative to some zero-shot methods, the performance gains are comparatively limited.

**Questions:**

1. There already exist numerous test-time adaptation (TTA) methods for OOD detection with strong results. Given the SOTA claim, a unified head-to-head comparison with representative methods such as AdaNeg \[1], OODD \[2], and AdaND \[3] is recommended.

2. Only AUROC is reported; please also include FPR95 and AUPR, which are more sensitive for open-set evaluation.

3. Appendix E.2 states that the attention complexity scales as $O(L^2)$ with the number of input samples L, which precludes large-scale data augmentation as in CoOp. When the number of shots k is large, a quantitative cost–benefit analysis is advised to assess the practical trade-off.

4. The current setup primarily uses CLIP-L/14 features(Table 2); please also evaluate RN50x16 and ViT-H/G to examine whether marginal gains diminish when the base model already provides strong separability.

\[1]AdaNeg: Adaptive Negative Proxy Guided OOD Detection with Vision-Language Models.

\[2]OODD: test-time out-of-distribution detection with dynamic dictionary.

\[3]Noisy Test-Time Adaptation in Vision-Language Models.

---

> ### Author Response · Authors · 2025-12-02
>
> [1] There already exist numerous test-time adaptation (TTA) methods for OOD detection with strong results. Given the SOTA claim, a unified head-to-head comparison with representative methods such as AdaNeg [1], OODD [2], and AdaND [3] is recommended.
>
> AdaNeg has better performance than ours as in their Table1. They claim even stronger performance then NegLabel. The authors acknowledge that our proposed few-shot method has worse performance than some zero-shot methods. The authors want to emphasize that the main contribution is the proposal of the concept of OOD learner and the demonstration of the concept -- the proposed method can be used to improve MCM and achieve a certain level of performance without any finetuning, though non-sota. OODD is orthoganal with us. As in OODD's section "7.1.IntegratewiththeCLIP-basedmethods", OODD can be added to MCM and NegLabel. All of AdaNeg, OODD, and AdaND are not few-shot OOD detection methods and not our direct baselines.
>
> [2] Only AUROC is reported; please also include FPR95 and AUPR, which are more sensitive for open-set evaluation.
>
> We appreciate the reviewer for this suggestion and will add those information to the appendix due to page limitation.
>
> [3] Appendix E.2 states that the attention complexity scales as O(L^2) with the number of input samples L, which precludes large-scale data augmentation as in CoOp. When the number of shots k is large, a quantitative cost–benefit analysis is advised to assess the practical trade-off.
>
> We tried to inference with 16-shot, the time cost is negligible due to the small size of OOD learner. When increasing to 1600-shot to mimic CoOp's image augmentation, we got Out Of Memory on 40G GPU.

---

### Official Review · Reviewer_FvYu · 2025-10-31

**Soundness:** 1
**Presentation:** 2
**Contribution:** 1
**Rating:** 2
**Confidence:** 3

**Summary:**

The work proposes an OOD detection approach based on attention module that claims to be inspired from the in-context learning using in GPTs. Unlike the prompt tuning approaches, the proposed method claims that they do not need fine-tuning which is a bit confusing in this interpretation and the presentation in the paper. The experiments are presented to compare the empirical results with the existing methods.

**Strengths:**

The paper introduces a nice and timely problem statement indicating that “prompt tuning lacks flexibility to new information, such as additional shots of images or new ID classes.”. This is indeed a drawback of many existing OOD detection baselines though they generally perform better than those derived from pre-trained models.

**Weaknesses:**

Weaknesses:

1.	The main issue is the ambiguity of the claim that “no fine-tuning” is needed. The authors are correct that no finetuning is needed on the encoders of the CLIP model during the text time. But, the attention module parameters are updated based on few-shot examples to learn the “in-context” or ID data. This is very similar to the prompt tuning methods such as CoOp, LoCoOp, SCT etc where the prompt vectors are learned using few-shot examples keeping the encoders frozen. Could you justify the reason why you still see your framework as “in-context” and compare with GPT-3?

2.	In addition, the methods also needs hyperparameter tuning and selection for $\lamda$ which is also against the claim of “in-context” learning and no fine-tuning.

3.	In the result table 1, more recent baselines like LoCoOp and SCT are not presented and FPR (False Positive Ratio) is also not reported. It is well-established in recent works that LoCoOp and SCT outperforms the vanilla CoOp by large margins. Table 2 also does not show the advantages of the approach. The results imply the proposed approach is not competitive enough with prompt-tuning based approaches.

**Questions:**

Please see Weaknesses.

---

> ### Author Response · Authors · 2025-12-02
>
> [1] The main issue is the ambiguity of the claim that “no fine-tuning” is needed. The authors are correct that no finetuning is needed on the encoders of the CLIP model during the text time. But, the attention module parameters are updated based on few-shot examples to learn the “in-context” or ID data. This is very similar to the prompt tuning methods such as CoOp, LoCoOp, SCT etc where the prompt vectors are learned using few-shot examples keeping the encoders frozen. Could you justify the reason why you still see your framework as “in-context” and compare with GPT-3?
>
> We believe this is the core misunderstanding on our paper. Consider the following: given a dataset named X with several classes and each class has 16-shot samples. CoOp will do prompt tuning on X. OOD learner will not do any finetuning on X. The attention module is pretrained on a large data, LAION 40M in this paper. OOD learner will simply inference the pretrained OOD learner with the few-shot samples without any finetuning on the few-shot samples.
>
> [2] In addition, the methods also needs hyperparameter tuning and selection for \lambda which is also against the claim of “in-context” learning and no fine-tuning.
>
> The authors ackowledge this weakness.
>
> [3] In the result table 1, more recent baselines like LoCoOp and SCT are not presented and FPR (False Positive Ratio) is also not reported. It is well-established in recent works that LoCoOp and SCT outperforms the vanilla CoOp by large margins. Table 2 also does not show the advantages of the approach. The results imply the proposed approach is not competitive enough with prompt-tuning based approaches.
>
> The authors acknowledge this weakness. The main contribution is the proposal of the concept of OOD learner and the demonstration of the concept -- the proposed method can be used to improve MCM and achieve a certain level of performance without any finetuning, though non-sota.

---

### Meta-Review · Area_Chair_KGMi · 2026-01-08

**Summary:**

Among the four reviewers, three gave negative scores and offered direct criticism in their reviews. They argue that the method has limited novelty and yields results that are comparable to many existing state-of-the-art approaches. They also note that the motivation and intended outcomes are not clearly articulated, and that the paper omits key data and analyses needed to support the claims. In addition, reviewers raise concerns that parts of the algorithm and theory appear inconsistent or not fully credible, and would require further refinement and clarification.

**Reviewer Concerns:**

In my view, the authors’ rebuttal is too brief to adequately address the reviewers’ concerns.

**Reviewer Scores:**

I do not think that the reviewers will change their scores.

---

### Decision · Program_Chairs · 2026-01-26

Reject